# Substrate diversity of NSUN enzymes and links of 5-methylcytosine to mRNA translation and turnover

Marco Guarnacci[1],*, Pei-Hong Zhang[2,3],*, Madhu Kanchi[1], Yu-Ting Hung[1], Hanrong Lin[1], Nikolay E Shirokikh[1], Li Yang[3], Thomas Preiss[1,4]

Maps of the RNA modification 5-methylcytosine (m$^5$C) often diverge markedly not only because of differences in detection methods, data depth and analysis pipelines but also biological factors. We re-analysed bisulfite RNA sequencing datasets from five human cell lines and seven tissues using a coherent m$^5$C site calling pipeline. With the resulting union list of 6,393 m$^5$C sites, we studied site distribution, enzymology, interaction with RNA-binding proteins and molecular function. We confirmed tRNA:m$^5$C methyltransferases NSUN2 and NSUN6 as the main mRNA m$^5$C "writers," but further showed that the rRNA:m$^5$C methyltransferase NSUN5 can also modify mRNA. Each enzyme recognises mRNA features that strongly resemble their canonical substrates. By analysing proximity between mRNA m$^5$C sites and footprints of RNA-binding proteins, we identified new candidates for functional interactions, including the RNA helicases DDX3X, involved in mRNA translation, and UPF1, an mRNA decay factor. We found that lack of NSUN2 in HeLa cells affected both steady-state levels of, and UPF1-binding to, target mRNAs. Our studies emphasise the emerging diversity of m$^5$C writers and readers and their effect on mRNA function.

## Introduction

Research into the distribution and function of mRNA modifications, often referred to as epitranscriptomics, has seen rapid growth in the past decade (Linder & Jaffrey, 2019). Although there were early indications of a sparse presence of modified nucleobases such as $N^6$-methyladenosine (m$^6$A) and 5-methylcytosine (m$^5$C) at internal positions within eukaryotic mRNA (Desrosiers et al, 1974; Perry & Kelley, 1974; Dubin & Taylor, 1975), the field was principally enabled by adaptations of high-throughput sequencing to map modified positions transcriptome-wide (Dominissini et al, 2012; Meyer et al,

2012; Squires et al, 2012; Linder & Jaffrey, 2019; Moshitch-Moshkovitz et al, 2022). Only a small subset of the many chemical marks known from classical RNA modification research (Boccaletto et al, 2022; Motorin & Helm, 2022) have thus far been explored in this way (Arzumanian et al, 2022), with m$^6$A and m$^5$C among those gaining the most acceptance for their roles in regulating cellular mRNA fate (Reid et al, 1999; Squires et al, 2012; Bohnsack et al, 2019; Trixl & Lusser, 2019; Schumann et al, 2020; Liu et al, 2021a; Chen et al, 2021; Wiener & Schwartz, 2021; Murakami & Jaffrey, 2022) and in organismic development and disease (He & He, 2021; Motorin & Helm, 2022; Guarnacci & Preiss, 2024).

m$^6$A is the most prevalent internal mRNA modification in eukaryotes and has been most extensively studied (He & He, 2021; Murakami & Jaffrey, 2022). Three main categories of proteins have been conceptualized to explain the occurrence and molecular roles of m$^6$A in mRNA metabolism, namely writers, readers, and erasers (Zaccara et al, 2019). A complex containing methyltransferase-like 3 (METTL3) as the catalytic subunit is the main dedicated mRNA m$^6$A writer. Specificity is largely achieved by recognising a (common) DRACH sequence motif combined with physical exclusion of the METTL3 complex from splice site regions (Uzonyi et al, 2023). Potential m$^6$A erasure from mRNA is catalysed by the m$^6$A demethylase AlkB homolog 5 (ALKBH5). Finally, YT521-B homology (YTH) domain proteins are the prevalent m$^6$A readers that directly bind and regulate m$^6$A-decorated mRNAs in several ways, but perhaps predominantly through affecting nuclear processing and cytoplasmic turnover (Zaccara et al, 2019; Murakami & Jaffrey, 2022).

Despite the early evidence of m$^5$C in mammalian mRNAs (Dubin & Taylor, 1975), no dedicated mRNA:m$^5$C methyltransferase (MTase) was identified, and research lay dormant in subsequent decades. This changed when coupling of RNA bisulfite treatment with short-read sequencing (bsRNA-seq) mapped multiple m$^5$C positions in human mRNA (Squires et al, 2012). With technological and bioinformatic refinements, this approach now suggests at least several hundred mRNA sites in transcriptomes from

[1]Shine-Dalgarno Centre for RNA Innovation, Division of Genome Science and Cancer, John Curtin School of Medical Research, Australian National University, Canberra, Australia [2]Shanghai Institute of Nutrition and Health, University of Chinese Academy of Sciences, Chinese Academy of Sciences, Shanghai, China [3]Center for Molecular Medicine, Children's Hospital of Fudan University, Shanghai Key Laboratory of Medical Epigenetics, International Laboratory of Medical Epigenetics and Metabolism, Institutes of Biomedical Sciences, Fudan University, Shanghai, China [4]Victor Chang Cardiac Research Institute, Sydney, Australia

Correspondence: Thomas.Preiss@anu.edu.au
*Marco Guarnacci and Pei-Hong Zhang contributed equally to this work

mammalian somatic cells or cell lines (Huang et al, 2019; Schumann et al, 2020; Hussain, 2021). Interestingly, site numbers can swell to several tens of thousands when analysing maternal mRNAs during early animal embryonic development (Liu et al, 2022). The responsible writers were identified among RNA:m$^5$C MTases of the NOL1/NOP2/SUN domain family (NSUN1-7), despite each of these already having established substrates, chiefly among either tRNAs or rRNAs (Reid et al, 1999; Bohnsack et al, 2019; Chen et al, 2021). The tRNA:m$^5$C MTases NSUN2 (Motorin & Grosjean, 1999; Brzezicha et al, 2006; Auxilien et al, 2012; Van Haute et al, 2019) and NSUN6 (Haag et al, 2015; Long et al, 2016; Liu et al, 2017) emerged as the main mRNA:m$^5$C writers, based on knockdown/KO experiments (Squires et al, 2012; Yang et al, 2017; Schumann et al, 2020; Liu et al, 2021a; Selmi et al, 2021). Accordingly, most mRNA m$^5$C sites fall into two categories: "type I" sites mimic the sequence and structural context around tRNA variable loops, a canonical substrate of NSUN2 (Huang et al, 2019; Schumann et al, 2020), whereas "type II" sites carry mixed tRNA-like features recognised by NSUN6 (Liu et al, 2021a; Selmi et al, 2021). There is also evidence that m$^5$C can be "erased" in mRNA, with oxidation by the ten–eleven translocation 2 (TET2) dioxygenase as the first step (Shen et al, 2018).

Three RNA-binding proteins (RBPs) were identified as m$^5$C readers, serine/arginine-rich splicing factor 2, Aly/REF export factor (ALYREF) and Y-box binding protein 1 (YBX1). Serine/arginine-rich splicing factor 2 was shown to regulate splicing of mRNA targets in chronic myeloid leukemia cells (Ma et al, 2023). ALYREF selectively promote target mRNA export from the nucleus (Yang et al, 2017), with recognised roles in controlling cell migration (Xu et al, 2020) and adipogenesis (Liu et al, 2021b). YBX1 preferentially binds m$^5$C-decorated mRNAs through its cold shock domain (Chen et al, 2019; Yang et al, 2019). In human urothelial carcinoma of the bladder, YBX1 recruits the ELAV-like 1 RBP to increase the stability of the mRNA for the oncogene heparin-binding growth factor, thereby promoting pathogenesis of bladder cancer (Chen et al, 2019). In early zebrafish embryos, YBX1 regulates the maternal-to-zygotic transition by promoting the stability of m$^5$C-containing maternal mRNAs through recruitment of the poly(A) binding protein cytoplasmic 1a (Yang et al, 2019). This resonates with intricate waves of maternal mRNA methylation mediated by NSUN2 and NSUN6, seen in both vertebrate and invertebrate species including flies, fish, frogs, mice, and humans (Liu et al, 2022). Finally, a broad inverse correlation between mRNA m$^5$C content and ribosome association was further seen in murine and human cells, consistent with a role of m$^5$C in repressing translation (Huang et al, 2019; Schumann et al, 2020; Liu et al, 2021a), perhaps through a yet unidentified reader protein.

Here, we derived a union list of 6,393 m$^5$C sites in the broader human transcriptome by re-analysing multiple published bsRNA-seq datasets. We used the statistical power inherent in this large number of sites to assess site features and determine the MTases responsible for m$^5$C deposition. We also analysed the proximity of m$^5$C sites to RBP binding sites. A functional interaction between one co-enriched RBP, UPF1, and m$^5$C in mRNA was investigated by performing RNAi-mediated UPF1 knockdown and CLIP experiments in wild-type and NSUN2 KO cells.

# Results

## A union list of m$^5$C sites in human transcriptomes

BsRNA-Seq data from seven human tissues (two replicates each) and five human cell lines (typically replicated; several conditions of altered NSUN gene expression) were collected from six published studies (Yang et al, 2017; Chen et al, 2019; Huang et al, 2019; Janin et al, 2019; Schumann et al, 2020; Liu et al, 2021a) (Table S1). These were re-analysed with a pipeline for m$^5$C site calling based on (Schumann et al, 2020) with minor modifications (see the Materials and Methods section). Briefly, after data pre-processing and mapping to the C-to-T and G-to-A converted genome (plus spike-in sequences as appropriate), non-conversion sites were called as variations by a custom script, for example, T-to-C in read2 and A-to-G in read1 (Fig 1A). Read distribution and overall cytosine conversion rate were checked to assess quality of the 50 datasets. For seven of the primary tissue samples, one of the two replicate datasets was excluded as metagene analysis showed strongly biased coverage toward mRNA 3′ ends, suggesting excessive degradation (Fig S1). Consistent with the original reports (e.g., [Schumann et al, 2020]), the remaining 43 datasets each showed good library complexity (Fig S2A) and high cytosine conversion rates, for example, ≥99.6% conversion rate for all annotated genes and ≥99.7% for protein coding genes (Table S1, Fig S2B, top panel). A set of filters was then applied to identify high-confidence m$^5$C candidate sites. Reads with more than three non-converted cytosines were discarded via a custom script to remove clustered non-conversion likely caused by RNA secondary structure (3C filter) (Fig S2B, bottom panel). Furthermore, we also excluded those non-converted cytosine positions for which less than 90% of reads passed the 3C filter (S/N90 filter). We then retained "C" positions with ≥20 reads coverage (20RC filter), ≥3 reads supporting the non-conversion, and with a non-conversion ratio ≥10% (10 MM). Sites that passed all filters in at least two biological replicates were called as high-confidence m$^5$C sites. Sites passing the filtering steps in only one of the replicates had to satisfy a higher threshold of ≥5 reads supporting non-conversion. After merging called sites from all datasets of non-manipulated NSUN gene expression, this yielded a union set of 6,393 m$^5$C sites (Table S2).

Concordance in site calls between samples will be limited by several factors. Expression levels of m$^5$C methyltransferase enzymes differ substantially across different tissues and cell lines (e.g., NSUN2/5/6; Fig S3A). Read coverage varies strongly between datasets for many sites (Fig S3B) because of differences in gene expression profile between samples and depth of library sequencing, which strongly reflects on the number of m$^5$C sites called in each samples of our union set (Fig S3C). Moreover, most sites show a non-conversion rate just above the empirically chosen ≥10% cut-off (Fig S3E), implying considerable random fluctuation as to whether a site is called in a given sample. Indeed, overlap in union sites called between different samples is modest (Fig S4A). In any given two sample comparison, most m$^5$C sites that are not called in common simply either did not reach the ≥20 read coverage threshold or exhibited stoichiometry below threshold (e.g., between 1–10%) in one of the samples (Fig S4B–D). Reassuringly, most union

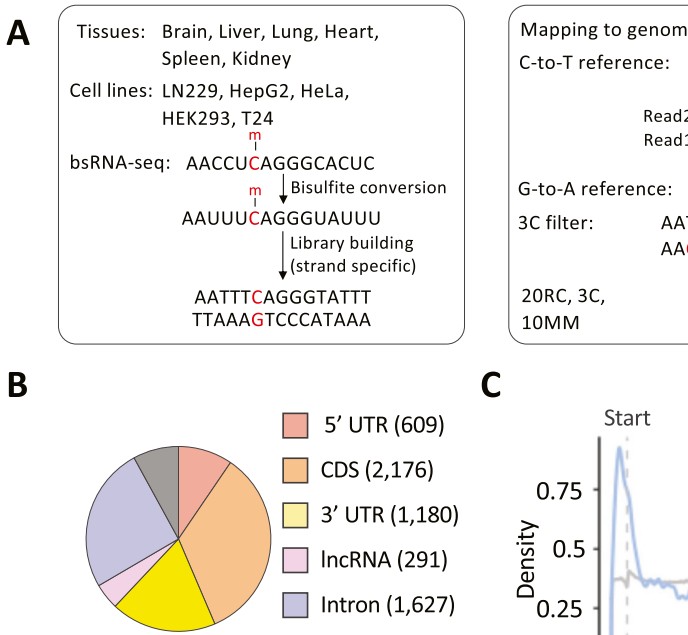

**A**

Tissues: Brain, Liver, Lung, Heart, Spleen, Kidney

Cell lines: LN229, HepG2, HeLa, HEK293, T24

bsRNA-seq: AACCU**C**AGGGCACUC
 ↓ Bisulfite conversion
 AAUUU**C**AGGGUAUUU
 ↓ Library building (strand specific)
 AATTT**C**AGGGTATTT
 TTAAA**G**TCCCATAAA

Mapping to genome:

C-to-T reference: AATTTTAGGGTATTT
 ↑ T-to-C
Read2 AATTT**C**AGGGTATTT
Read1 TTAAA**G**TCCCATAAA
 ↓ A-to-G
G-to-A reference: TTAAAATCCCATAAA

3C filter: AATTT**C**AGGGTATTT
 AA**CCT C**AGGGTA**C**TT

20RC, 3C, 10MM

**B**

5' UTR (609)
CDS (2,176)
3' UTR (1,180)
lncRNA (291)
Intron (1,627)
Intergenic (510)

Number of total m5C sites = 6,393

**C**

Start Stop

m5C (3,965)
C (3,377,426)

Density

5' UTR CDS 3' UTR

**Figure 1. Generation of a union list of human m5C.**
**(A)** Schematic of the strategy used for generating a union set of human m5C sites: selection of publicly available bsRNA-Seq data from six human tissue and five cell lines and high confidence m5C sites calling with stringent and unified pipeline. **(B)** Distribution of 6,393 m5C sites from union set across different RNA species. Site annotation was performed according to the transcripts as recorded in the hg38 GENCODE v32 annotation (UCSC). **(C)** Metagene density plot showing the distribution of m5C sites along mature mRNAs. mRNA regions were scaled to the mean of their respective length (5'UTR: 270 nt; CDS: 2,058 nt; 3'UTR: 1,817 nt). m5C distribution is shown in blue, background cytosine distribution in grey.

set sites are found in protein coding genes (Fig 1B) and they distribute nonrandomly along mature mRNAs (3,965 sites), showing enrichment around the start codon (Fig 1C), a pattern found repeatedly with single source material in published work (Yang et al, 2017; Chen et al, 2019; Huang et al, 2019; Schumann et al, 2020) and again here (Fig S4E). This establishes utility of our union set as a larger yet still prototypical list of human transcriptomic sites across multiple cell types for further exploration of m5C enzymology and function.

## The spectrum of mRNA m5C writer enzymes

We extended our analyses to several datasets derived from cell lines depleted of NSUN2 (HeLa human cervical cancer cells; [Yang et al, 2017; Huang et al, 2019]) or NSUN6 (HEK293T human embryonic kidney cells; [Liu et al, 2021a]) (Table S3). By re-analysing NSUN2 KO or knockdown data from HeLa cells, we found that 798 m5C sites (91.3%) in Huang et al (2019) and 859 (52.8%) in Yang et al (2017) had a reduced non-conversion level upon NUSN2 depletion compared with wild-type control and were therefore regarded as NSUN2-dependent. Similarly we found that 275 (86.7%) sites had reduced methylation level in HEK293T cells upon NSUN6 KO (Liu et al, 2021a). Lastly, upon re-analysis of NSUN2 and NSUN6 double KO data (Liu et al, 2021a) we observed the m5C stoichiometry of 44 (8.19%) remained unaffected. In this way, we identified 1,368 NSUN2-mediated, 275 NSUN6-mediated and 44 NSUN2/6 independent m5C sites using experimental data. As expected, we found a predominance of sequence and structural features of type I and type II m5C sites, respectively (Liu et al, 2021a). Type I sites are characterized by a G-rich triplet downstream of the modified position at the 5' end of a hairpin (Fig 2A and B—top) (Huang et al, 2019;

Schumann et al, 2020), resembling the context of m5C sites (C48-50) modified by NSUN2 in the variable loop of multiple tRNAs (Auxilien et al, 2012). Type II sites are adjacent to a 3' UCCA motif and located in the loop of a hairpin (Fig 2A and B–middle) (Liu et al, 2021a; Selmi et al, 2021). In this way they comply with a mix of sequence and shape-selective substrate recognition by NSUN6, which otherwise targets C72 next to a UCCA motif at the 3' end of some tRNAs (Long et al, 2016; Liu et al, 2017). Based on the sequence characteristics at position 1 to 5 downstream of these experimentally identified NSUN2 and -6 targets, we generated a position weight matrix (PWM). We then predicted the enzymes responsible for m5C sites in our union set (Fig 2C) and individual samples (Fig S3D) using the PWM via FIMO software (Grant et al, 2011). This indicated that in the wider human transcriptome, around two-third of m5C sites are likely deposited by NSUN2, whereas around one-sixth are deposited by NSUN6 (Fig 2C) and these sites still comply with type I and type II structural characteristics, respectively (Fig S4F and G). The remaining one-sixth of sites that could not be attributed to either NSUN2 or NSUN6 showed no strong discernible commonalities (Fig 2A and B–bottom; Fig S4F and G–bottom). Given that MTase expression levels will vary between cell and tissue sources (e.g., Fig S3A), we tested for any co-variation of site methylation levels. Indeed, we found a selective positive correlation of the methylation level of NSUN2- or NSUN6-dependent sites with NSUN2 or NSUN6 expression levels, respectively. NSUN2/6-independent sites ("other") showed no correlation with expression levels of any of the NSUN MTases tested (Fig 2D). This confirms and extends expectations that NSUN2 and -6 are the main m5C writers in mRNA but also leaves scope for other enzymes that might deposit m5C sites on mRNA.

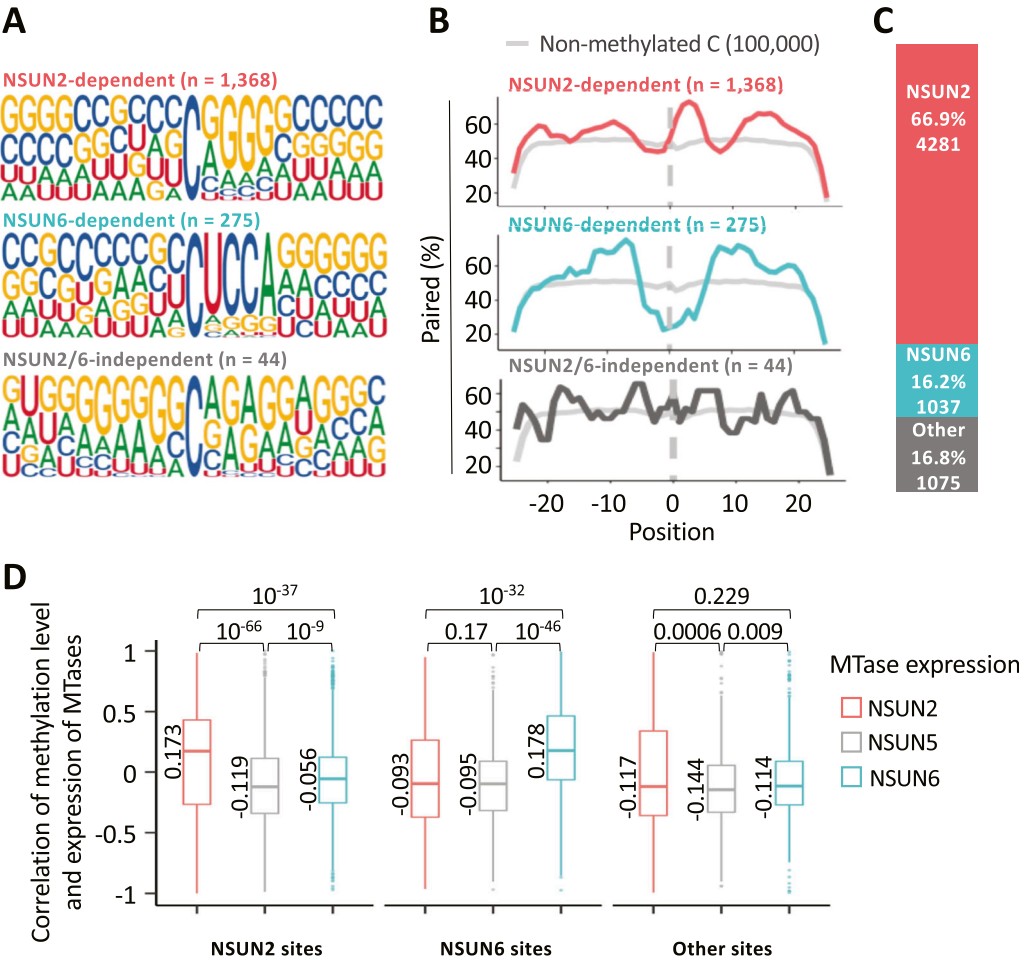

**Figure 2. Widespread NSUN2 and NSUN6-dependent m⁵C methylation.**
**(A)** Consensus motifs surrounding m⁵C sites (shown as central "C") assigned to either NSUN2 (top), NSUN6 (middle) or "other" based on experimental NSUN depletion data. The number of m⁵C sites used for the generation of each sequence logo is indicated in the figure. **(B)** Base-pairing propensity meta-profile of regions surrounding m⁵C sites (position "0") deposited by either NSUN2 (top), NSUN6 (middle) or "other" according to experimental data. 100,000 non-metylated random cytosines were used as background control (in grey). **(A, C)** Sequence logos from (A) were used to predict NSUN-dependence of m⁵C sites in the wider union set. **(D)** Pearson correlation coefficient between m⁵C stoichiometry and methytransferases expression for each site in the union set with more than 20 read coverage in at least five samples. *P*-value was calculated by *t* test.

We next looked at bsRNA-seq data from LN229 human glioblastoma cells, where NSUN5 was either epigenetically silenced (control) or overexpressed by lentiviral transduction (OE) (Janin et al, 2019). NSUN5 is a rRNA:m⁵C MTase responsible for modifying C: 3782 in human 28S rRNA and equivalent positions in other organisms (Sharma et al, 2013; Schosserer et al, 2015; Heissenberger et al, 2019; Janin et al, 2019). We found 446 mRNA m⁵C sites in the control but 3,921 sites in NSUN5 OE (Fig 3A), in the context of similar read coverage (Fig S5). Sites in the control condition complied with type I sequence features and were enriched around start codons (Fig 3B). By contrast, in the NSUN5 OE condition, a GUNGCCANNUG motif was found to be prevalent, with broader m⁵C site enrichment along the mRNA coding sequence instead (Fig 3B). This sequence motif clearly resembles the sequence context on human 28S rRNA (C:3782) (Fig 3C), which is highly conserved from yeast to human (Heissenberger et al, 2019). Using the established PWM of NSUN2 and -6 targets (details in the Materials and Methods section), we

predicted sites in LN229 cells to be either NSUN2 or -6 dependent (Fig S5). Looking specifically at sites determined to be NSUN5-dependent, based on their increased non-conversion levels upon NSUN5 overexpression (Fig S5, left - "NSUN5-dependent"), we found a narrow but specific pattern of predicted secondary structure (Fig 3D), which is consistent with the predicted structure in Liu et al (2023), and reminiscent of the situation of 28S rRNA C:3782, which is in a relatively unstructured region but immediately downstream of two base-paired positions (Fig 3E).

The NSUN5-targeted sequence motif described above has also just been identified in bsRNA-seq data from early developmental stage samples as well as Nocodazole-treated HeLa cells (Noc-HeLa), and termed "type III", using a novel computational framework for epitranscriptomic motif discovery (Liu et al, 2023). Nocodazole arrests cells in mitosis when most nuclear content is released into the cytoplasm, providing more opportunity for predominantly nuclear/nucleolar MTases such as NSUN5 to modify

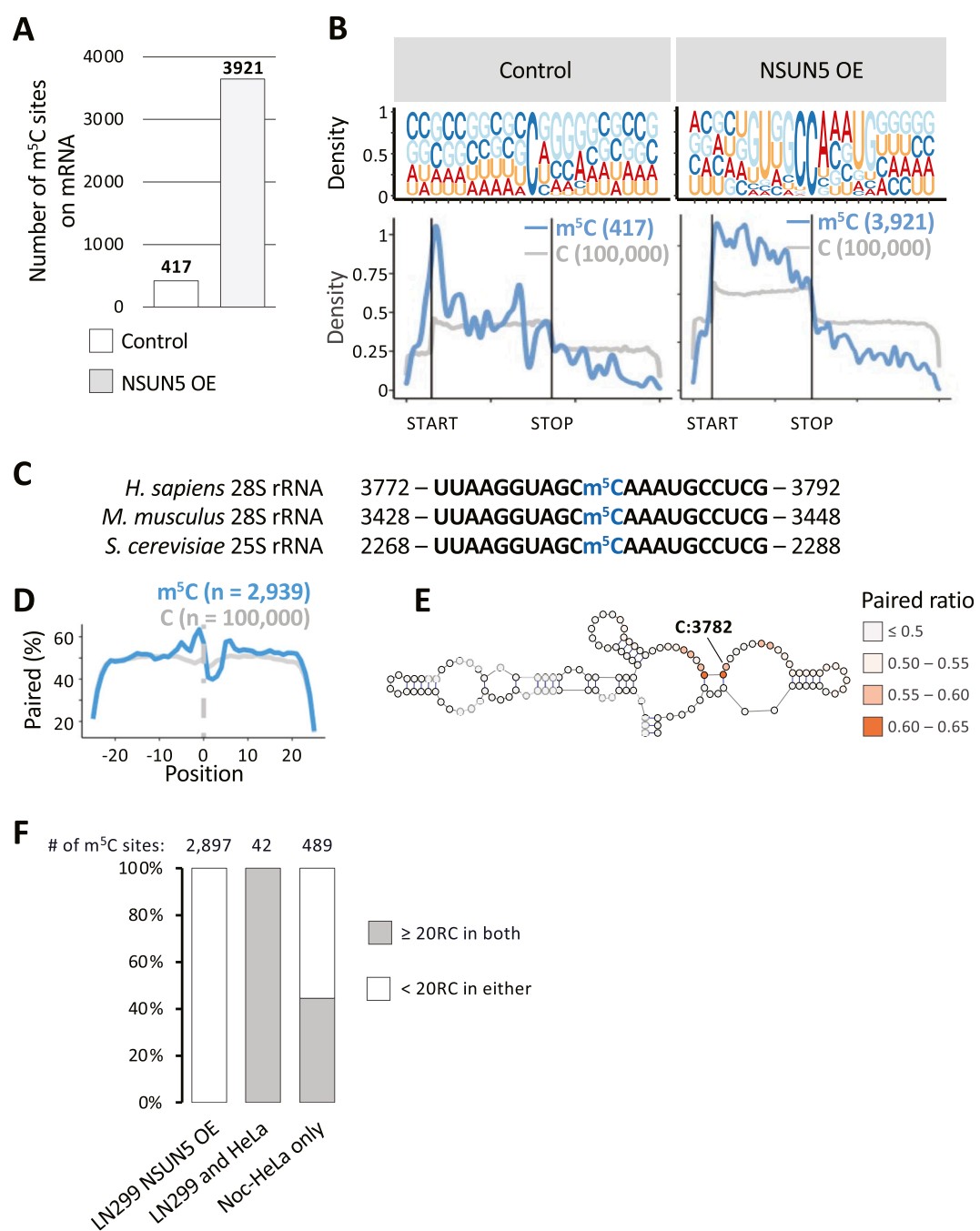

**Figure 3. NSUN5 deposits m⁵C modification on mRNA.**
(A) Number of m⁵C found on mRNA in LN229 cells with epigenetically silenced (control) or overexpressed (OE) NSUN5. (B) Consensus motif surrounding m⁵C sites (top) and m⁵C distribution over mRNA (bottom), in control (left), and OE LN229 cells (right), plotted as in Fig 1C and D. (C) Alignment of 28S rRNA sequence carrying an m⁵C modification (in blue) across three different species. (D) Base-pairing propensity meta-profile around NSUN5-dependent m⁵C sites, as in Fig 1E. (D, E) Base-pairing percentage of the region ±20 nt surrounding m⁵C sites from panel (D) are displayed within the structure of 28S rRNA, aligning candidate sites with the NSUN5-dependent m⁵C:3782 position of human 28S rRNA. (F) Stacked bar chart showing NSUN5-dependent m⁵C mRNA sites detected in LN299 NSUN5 OE cells only left; (Janin et al, 2019), Noc-HeLa cells only right; (Liu et al, 2023), or both conditions (middle). Grey shading indicates the percentage of sites that have coverage (≥20RC) in both samples.

mRNA (Liu et al, 2022). Comparison of sites between the two cell lines/conditions was severely limited by differences in gene expression. 42 m⁵C sites were called for NSUN5 in both conditions (Fig 3F). Remarkably, these were all the sites among 2,939 in LN229 OE that had sufficient coverage (≥20RC) also in data from Noc-HeLa. Conversely, whereas around half of the 531 NSUN5 sites called in Noc-HeLa also had coverage in LN229 OE, 42 still represented a substantial overlap. Altogether, this further substantiates that NSUN5 is another MTase capable of depositing m⁵C on mRNA.

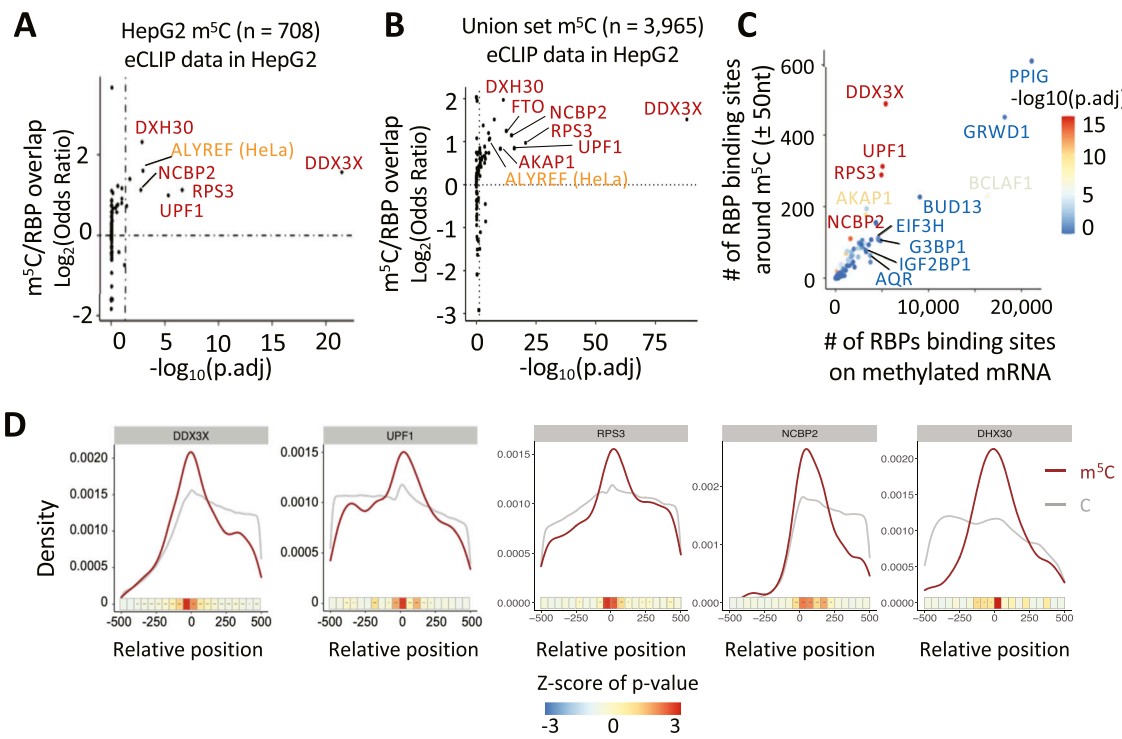

**Figure 4. Multiple RBP binding sites are enriched near m⁵C sites.**
**(A, B)** Chi-Squared Test showing the enrichment between RBP binding sites from eCLIP data in HepG2 cells and m⁵C sites (±50 nt) on mature mRNA from HepG2 cell line with (B) Chi-Squared Test showing the enrichment between RBP binding sites from eCLIP data in HepG2 cells and m⁵C sites (±50 nt) on mature mRNA from our union set. **(A, B)** Highlighted in red are the five enriched RBPs consistently found in both comparisons. ALYREF RBP (in orange) was used as a positive control. *P*-values were determined using Chi-Squared Test and "fdr" adjustment. **(C)** Chi-Squared Test showing the number of RBP binding sites (eCLIP data from HepG2 cells) overlapping with m⁵C (±50 nt) on mRNAs in HeLa cells, compared with the number of RBP binding sites in methylated mRNA (>50 nt). *P*-values were determined using Chi-Squared Test and "fdr" adjustment. **(A, B, D)** Metagene density plot showing the distribution of m⁵C sites on mRNAs targeted by top enriched RBPs (in red) form (A, B). Centred in position 0 is the middle point of the RBP's footprint on said mRNAs. The density of non-methylated cytosines from the same mRNAs was used as background control.

## RBP binding sites are enriched around m⁵C sites

To investigate potential spatial proximity between m⁵C sites and footprints of known RBPs we obtained enhanced UV cross-linking and immunoprecipitation (eCLIP) sequencing datasets from EN-CODE, available for 122 RBPs from the human lymphoblast K562 cell line and for 104 RBPs from human liver cancer HepG2 cells (Van Nostrand et al, 2020). We further used iCLIP data for the known m⁵C reader ALYREF in HeLa cells from CLIPdb as positive control (Yang et al, 2015). It is reported that genes with comparable expression levels in different cell lines show similar RBP enrichment peaks (Van Nostrand et al, 2020), justifying our approach to compare eCLIP and m⁵C data from different sources. Subsequently, we performed enrichment analysis of RBPs binding near m⁵C site deposition in mature mRNAs. After identifying the RBP binding sites as footprint centre positions, we established for each RBP, sets of mRNAs that featured both, m⁵C sites (from the union set or selected subsets) and footprints of said RBP (from both, K562 and HepG2 cells, or individually; see the Materials and Methods section for details). RBP-adjacent regions around each RBP's footprint centre (±20/30/50/70 nucleotides) were defined and the distribution of m⁵C and unmodified cytosine positions assessed within and outside these regions. This approach found statistically significant m⁵C enrichment in several RBP-adjacent regions that often persisted

irrespective of interval width and dataset cross-comparison (Fig S6A–D). We settled on a ±50 nucleotide interval around RBP footprint centres for further analyses. Using the same method, ALYREF was also seen as enriched near m⁵C sites in HeLa cells (Fig S6E), indicating the reliability of this method. To test the robustness of the association between RBP and m⁵C, we compared the enrichment of HepG2 RBP footprints with m⁵C sites in mRNA from different cell contexts, including HepG2-derived (Liu et al, 2021a) and HeLa-derived sites (Yang et al, 2017; Huang et al, 2019; Schumann et al, 2020) (Fig S6E) and the full union set (Fig 4B). When using 708 HepG2-derived m⁵C sites, we identified m⁵C co-enrichment with five RBPs: DDX3X (DEAD-box helicase 3 X-linked), UPF1 (Up-frameshift protein 1), NCBP2 (nuclear cap-binding protein subunit 2), RPS3 (ribosomal protein S3), and DHX30 (DExH-box helicase 30) (Fig 4A). Using the 2,342 m⁵C sites derived from HeLa cells (Fig S6E) or the 3,965 mRNA sites from our union set (Fig 4B) yielded seven RBPs enriched in both comparisons, the five seen before plus FTO (Fat mass and obesity associated) and AKAP1 (A-kinase anchoring protein 1). We also repeated the analysis using eCLIP data from K562 cells and found six of the seven RBPs to co-enrich with m⁵C sites from the union set (lack of data for AKAP1 in K562 cells precluded its assessment; Fig S6F).

We performed several additional analyses to further characterize the m⁵C-RBP associations. First, we plotted footprint

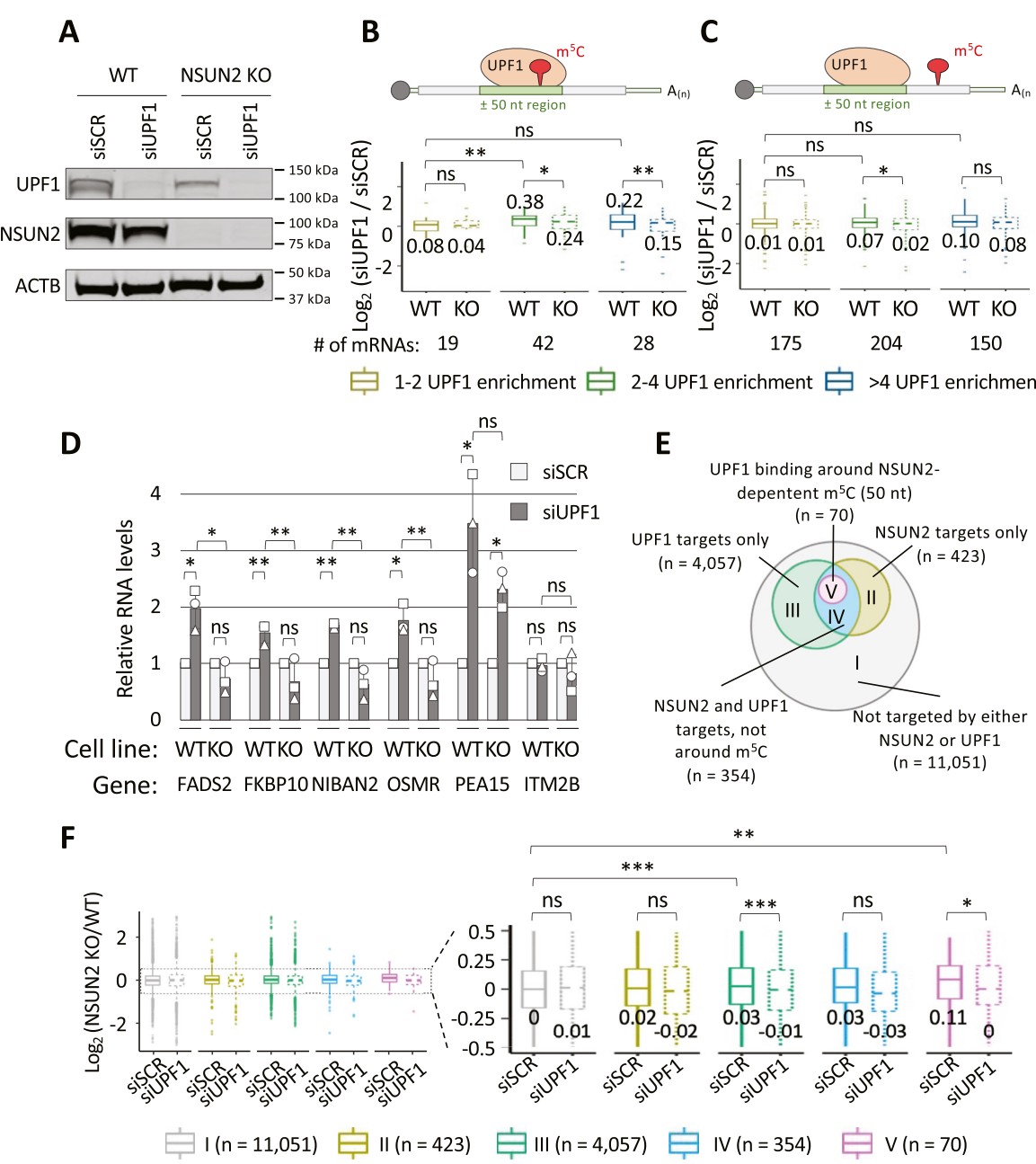

**Figure 5. UPF1 function is affected by the lack of NSUN2.**
**(A)** Western blot of extracts from wild-type and NSUN2 KO HeLa cells after siRNA-mediated UPF1 knockdown (siUPF1). A scrambled siRNA (siSCR) was used as control. Shown are probings for UPF1, NSUN2; ACTB was used as loading control. A representative image from replicated experiments (N = 3) is shown. **(B, C)** Box plot showing the relative RNA levels (measured by RNA-seq) of UPF1 targets with $m^5C$ sites located within (B) or outside (C) a ±50 nt interval around the mid-point of UPF1 footprints, upon UPF1 knockdown in WT (solid line), and NSUN2 KO (dashed line) HeLa cells. mRNAs groups are colour-coded depending on their enrichment level with UPF1 protein. *P*-values were calculated using *t* test. **(B, D)** RT-qPCR measurement of UPF1 mRNA target levels (selected from panel (B)), upon UPF1 knockdown in WT and NSUN2 KO HeLa cells. PEA15 and ITM2B were used as non-methylated, positive and negative controls for UPF1 binding, respectively. Values were normalised over ACTB and "siUPF1" values expressed relative to "siSCR" treatment set as 1. *P*-values were calculated using two-tailed *t* test. *t* test-derived *P*-values: **P* < 0.05; ***P* < 0.01; ****P* < 0.001. N = 3. **(E)** The HeLa mRNA transcriptome was subdivided into five categories (Roman numerals) and overlapped, based on UPF1 binding, $m^5C$ sites targeted by NSUN2, and distance between the two. **(F)** Box plot showing RNA levels (measured by RNA-seq) in NSUN2 KO and WT HeLa cells in control (siSCR, solid line) and UPF1 knockdown (siUPF1, dashed line) conditions. **(E)** RNA groupings and colour-coding are defined in the Venn diagram in panel (E). *P*-values were calculated using *t* test.

counts of RBPs (from Fig 4A; HepG2 eCLIP data) inside and outside a ±50 nucleotide interval around $m^5C$ sites (HeLa cell data). This further confirmed a positive association of footprints with $m^5C$ for five RBPs, DDX3X, UPF1, RPS3, NCBP2, and AKAP1 (Fig

4C). Second, we plotted the density of $m^5C$ and unmodified cytosines relative to the centre of footprints for the five most consistently enriched RBPs (Fig 4D), again showing marked co-enrichment.

Through their recognised functions, the seven enriched RBPs associate m⁵C sites with several mRNA-related processes. For example, NCBP2 promotes mRNA nuclear export together with ALYREF (Kataoka, 2023), a previously known m⁵C reader (Yang et al, 2017). RPS3, AKAP1, DHX30, and DDX3X all have roles in mRNA translation (Dong et al, 2017; Bosco et al, 2021; Ryan & Schröder, 2022; Bohnsack et al, 2023; Cohen et al, 2024). FTO is a demethylase involved in m⁶A metabolism (Mauer et al, 2017). Finally, UPF1 is involved in several mRNA decay pathways (Lavysh & Neu-Yilik, 2020).

**UPF1 function is affected by the lack of NSUN2**

We chose UPF1 for experimental follow-up to investigate a functional association with m⁵C. siRNA-mediated UPF1 knockdown was performed in both WT and NSUN2 KO HeLa cells (Fig 5A), to compare the effects of the lack of UPF1 with and without NSUN2-mediated mRNA methylation. Details on the generation of NSUN2 KO cell lines are described in Acera Mateos et al (2024). We performed RNA sequencing for three biological replicates of the RNA samples derived from UPF1 knockdown experiment (Fig S7A and B). Compared with the scrambled siRNA (siSCR) control condition, we found that the knockdown of UPF1 in WT and NSUN2 KO cells resulted in a similar number of down-regulated genes (~2,000). However, the number of up-regulated genes was lower in NSUN2 KO (1,908) compared with WT (2,365) cells (Fig S7C). To investigate whether this differential regulation could be attributed to the different m⁵C content in mRNA in WT versus NSUN2 KO cells, we used bisulfite sequencing data of WT untreated cells and NSUN2 KO/KD HeLa cells (Yang et al, 2017; Huang et al, 2019) to compile a list of 1,633 NSUN2-dependent m⁵C sites in HeLa cells (Table S1). We then selected those m⁵C sites located on mRNA that are also enriched in UPF1 binding and looked at their expression levels upon UPF1 knockdown, in WT versus NSUN2 KO cells. Those mRNA with an m⁵C modification overlapping with UPF1 binding site ("in region"—within ±50 nucleotides) were found to significantly accumulate in WT cells upon UPF1 knockdown; this up-regulation is significantly reduced when UPF1 is knocked down in NSUN2 KO cells (Figs 5B and S7D). Interestingly, this differential regulation between WT and NSUN2 KO cells was either lost, or strongly reduced, for those mRNAs whose m⁵C sites do not overlap with UPF1 binding site ("not in region"—outside ±50 nucleotides) (Fig 5C). As an additional control, we looked at the RNA levels of those mRNAs with UPF1 binding overlapping with NSUN2-independent m⁵C sites. In particular, these sites have the m⁵CUCCA consensus motif associated with NSUN6 methyltransferase and showed no reduction in methylation upon NSUN2 KO/KD (Liu et al, 2021a). As expected, the down-regulation of UPF1 protein had the same regulatory effect on these mRNAs in both WT and NSUN2 KO (Fig S7E). RT-qPCR validation of four mRNAs selected from Fig 5B confirmed a significant accumulation of RNA upon UPF1 knockdown in WT cells and the lack of such regulation upon UPF1 knockdown in NSUN2 KO HeLa cells (Fig 5D).

We then checked the effects of the lack of NSUN2 on the steady state of mRNAs. We divided the transcriptome into five groups (Fig 5E): non-targets (I); NSUN2 targets only (II); UPF1 targets only (III); UPF1 and NSUN2 targets (IV)—same as Fig 5C; UPF1 binding around NSUN2-dependent m⁵C (V)—same as Fig 5B. When comparing the RNA levels of each group upon NSUN2 KO, with (siSCR) and without

(siUPF1) UPF1 protein, we found that group V had the highest difference in RNA level relative to group I (Fig 5F, right panel), which recapitulates the effects of UPF1 knockdown and links the effects of UPF1 protein with the methyltransferase activity of NSUN2.

We then decided to assess whether the reduced methylation content in mRNA caused by the KO of NSUN2 also affects the binding ability of UPF1 protein. To test this, we performed UPF1 CLIP experiments in five biological replicates of both WT and NSUN2 KO HeLa cells (Fig 6A). RT-qPCR analysis on the RNA immunoprecipitated with UPF1 protein revealed a significant increase in binding activity for UPF1 upon NSUN2 KO. In particular, the same genes tested in UPF1 KD experiment (Fig 5D) were found to be significantly more enriched in NSUN2 KO samples compared with their WT control (Fig 6B and C).

Collectively, these results indicate that removing nearby m⁵C sites negatively affected UPF1's capacity to promote target mRNA degradation, when improving its ability to persistently bind to those targets.

# Discussion

Using an aggregated list of transcriptomic m⁵C sites, this study confirmed NSUN2 and 6 as the main m⁵C "writers" for mRNAs in human somatic cells and cell lines and furthermore showed that NSUN5 can also act on mRNA. mRNAs sites targeted by each NSUN variant mimic the features that the respective enzyme would also recognise in its canonical tRNA or rRNA substrates. m⁵C sites in mRNA were found to be robustly enriched around footprints of several RBPs. The molecular functions of these RBPs reinforced expectations of an involvement of m⁵C in mRNA nuclear export, translation, and turnover. The function of one of these RBPs, UPF1, was shown to be affected in cells lacking NSUN2.

This work adds to the emerging picture that, beyond NSUN2 and NSUN6, several other NSUN MTases can and will modify RNA outside their canonical substrate range, including mRNA, given the right conditions. Our observations here on NSUN5 are consistent with a recent study that identified both NSUN5 and NSUN1 as MTases with a broader substrate range in vertebrates (Liu et al, 2023). NSUN7-dependent modification of mRNA substrates was further recently described in human liver (Ortiz-Barahona et al, 2023).

The first condition to be met is that the noncanonical RNA target contains sequence and/or structural features that are specifically recognised by the enzyme in question. Rui Zhang and colleagues have proposed a numbered motif nomenclature for sites in the broader transcriptome to explain this for NSUN2 (type I), NSUN6 (type II), NSUN5 (type III), and NSUN1 (type IV) (Liu et al, 2023). The canonical substrate mimicry is perhaps best understood for type II sites, which form a hairpin presenting the modified position and 3′ adjacent sequences (CUCCA) in the loop. A human NSUN6-tRNA co-crystal structure is available and it shows marked conformational changes in the acceptor branch of the tRNA to fit the tRNA 3′ end into a cleft formed between the PUA and MTase domains of the enzyme, allowing recognition of the U73 by the RRM motif and the CCA end by the PUA domain when exposing the C72 position to the catalytic site. The PUA domain further makes specific contacts

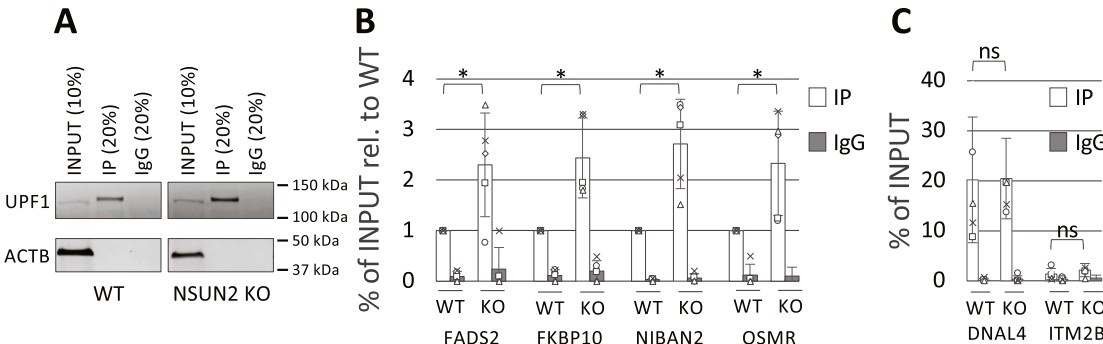

**Figure 6. UPF1 binding is affected by the lack of NSUN2.**
**(A)** Western blot to assess success of UPF1 CLIP experiments in WT and NSUN2 KO HeLa cells. 10% of cell extracts (input) and 20% of anti-UPF1 immunoprecipitation and a non-specific IgG control were loaded and probed for UPF1 and ACTB as loading control. A representative image from replicated experiments (N = 5) is shown. **(B, C)** RNA extracted from samples in panel A were analysed by RT–qPCR to assess enrichment levels of the same mRNA candidates as in Fig 4D with UPF1 in WT and NSUN2 KO HeLa cells. **(B, C)** Values were normalised over RNA SPIKE-IN and expressed as percentage of input relative to WT (B) or percentage of input (C). DNAL4 and ITM2B were used as positive and negative controls, respectively. $P$-values were calculated using two-tailed $t$ test. $t$ test-derived $P$-values: *$P < 0.05$; **$P < 0.01$; ***$P < 0.001$. N = 5.

with the D-stem of the tRNA (Liu et al, 2017). Mutation analyses and modelling of type II sites into the NSUN6 structure then showed how the hairpin mimics key portions of the 3D tRNA structure, whereas the loop appropriately offers the CUCCA motif to the enzyme (Liu et al, 2021a). Our current understanding of type III sites suggests that NSUN5 primarily recognises an extended sequence motif GUNGCCANNUG that matches the highly conserved context of C:3782 in 28 s rRNA (this study and Liu et al [2023]). An unresolved question is whether there are additional constraints that guide enzyme preference for certain regions of mRNA. For example, what guides the enrichment of NSUN2-modified sites in the 5′ region of mRNAs (Liu et al, 2023), and NSUN6 sites in the 3′ region of mRNAs (Selmi et al, 2021)?

The second condition is that the enzyme and substrate must be able to physically interact within the cell. This will be affected by the expression pattern of *NSUN* genes, which are known to be regulated in a developmental and tissue-selective manner, e.g., high during murine development and in the brain (Chi & Delgado-Olguín, 2013; Bohnsack et al, 2019), and dysregulated in tumours, e.g., (Frye & Watt, 2006; Janin et al, 2019; Jiang et al, 2020; Ortiz-Barahona et al, 2023). The distinct intracellular localisations of NSUN proteins are likely to be a major factor, with some residing in the nucleus (NSUN2, NSUN7), others in the nucleolus (NSUN1, NSUN2, NSUN5), mitochondria (NSUN3, NSUN4) and cytoplasm (NSUN6) (Bohnsack et al, 2019). Nucleolar enzymes might only interact fleetingly with mRNA, unless they are released from this compartment. High overexpression of NSUN5 as analysed here likely led to broader cellular distribution of the enzyme causing a large increase in detectably modified type III sites. Nevertheless, this likely still reflects physiological phenomena as type III/NSUN5 and type IV/NSUN1 sites were prominently found in nocodazole-treated HeLa cells (Liu et al, 2023). Nocodazole arrests cells in mitosis after nucleolar and nuclear envelope breakdown, thus prolonging a phase of the cell cycle that allows release of nucleolar content into the broader cellular environment. Type III/NSUN5 sites were also present in data from human oocytes, coinciding with similar nuclear content release (Liu et al, 2023). Maternal mRNAs are massively modified with m5C, which is required for proper

maternal-to-zygotic transition and this involves NSUN2, NSUN6 (Liu et al, 2022), and NSUN5 (Ding et al, 2022). These emerging links to cell cycle and development suggest that other (patho-)physiological contexts might exist in which m5C modification of mRNAs is particularly common and relevant.

Several leads regarding m5C function in mRNA have emerged here from its persistent co-enrichment with the footprints of certain RBPs. The research presented in this study did not assess whether any of these RBPs are direct m5C reader proteins; some of them might be but for others an alternative explanation is more likely. One of them, FTO, preferentially demethylates *N*6,2′-*O*-dimethyladenosine in the 5′ cap structure of mRNA affecting stability (Mauer et al, 2017), thus potentially linking m5C with m6A metabolism. An interesting Candidate is NCBP2, a subunit of the nuclear cap-binding complex that promotes multiple steps in mRNA metabolism, including nuclear export together with ALYREF (Kataoka, 2023). ALYREF reportedly is a direct m5C reader protein that promotes export of bound mRNA (Yang et al, 2017); the proximity of NCBP2 footprints to mRNA m5C sites is thus likely indirect but further implicates m5C in mRNA export. Cap-binding complex also has a role in mRNA translation as have several other enriched RBPs. RPS3 forms part of the entry channel of the 40S ribosomal subunit where it promotes mRNA binding (Dong et al, 2017), whereas AKAP1 is a multivalent protein with roles also in binding mRNA for local protein synthesis at the mitochondrial outer membrane (Cohen et al, 2024). Three RNA helicases (Bohnsack et al, 2023), DHX30, DDX3X and UPF1, were also persistently enriched. Two of these again link to translation; a cytoplasmic form of DHX30 is implicated in coordinating cytoplasmic translation and mitochondrial function (Bosco et al, 2021). DDX3X is strongly linked to the initiation stage of translation (Ryan & Schröder, 2022) and, interestingly, was shown to bind the NRAS mRNA 5′UTR in a similar manner to NSUN5 (Herdy et al, 2018). Thus, there are now three leads to potentially explain the known connection between m5C and mRNA translation (Huang et al, 2019; Schumann et al, 2020; Liu et al, 2021a). The link we chose to experimentally follow up here, however, is to mRNA turnover. The ATP-dependent RNA helicase UPF1 is involved in nonsense-mediated decay—which degrades

mRNAs with premature stop codons but also error-free transcripts (Kurosaki et al, 2019; Karousis & Mühlemann, 2022)—and in several other mRNA decay pathways (Lavysh & Neu-Yilik, 2020). Whereas targets/triggers, effectors, and level of available mechanistic insight vary greatly between these UPF1-dependent pathways, they often are translation-dependent and they typically require intact ATPase and helicase functions and hyperphosphorylation of the protein (Lavysh & Neu-Yilik, 2020). Notably, UPF1 can clamp onto RNA in an inactive state and its activation to a highly processed helicase requires conformational changes (Franks et al, 2010; Chakrabarti et al, 2011). In the context of NMD and some of the other pathways, these changes are induced by UPF2 binding and SMG1 (suppressor with morphogenetic effect on genitalia)-mediated phosphorylation (Lavysh & Neu-Yilik, 2020; Karousis & Mühlemann, 2022). These features of UPF1 function offer a plausible explanation for the observations made here, namely that bound UPF1-proximal m5C sites might somehow favour switching of UPF1 into its active state, simultaneously decreasing its stable RNA-binding and enhancing its ability to promote mRNA decay. In this scenario, UPF1 is likely not a direct reader of m5C; instead the modification might affect local RNA structure and/or alter binding of other decay factors that then effect UPF1 conformational change. The validity of this hypothesis and the underlying mechanism for such action should be addressed in future work.

## Materials and Methods

### Quality control and mapping of RNA BS-seq reads

For each public dataset, raw reads were subjected to FastQC (v0.11.9). Low-quality bases and adaptor sequences were removed using Trimmomatic (v0.38) with options ILLUMINCLIP:Adapter.fa:2:30:10:8:true LEADING:3 TRAILING:3 SLIDINGWINDOW: 4:20 MINLEN:50. We checked the conversion level for each dataset using spike-in sequence (Fig S2B–top, Table S1). Clean reads were mapped on the 45S rRNA, pre-tRNA, and predicted mature tRNA using meRanT tool (align bsRNA-seq reads to reference using Bowtie2 v2.3.5) in MeRanTK (v1.2.1b) with options (−k 10). Then, paired reads were obtained from unmapped reads using seqkit. Finally, unmapped reads pairs were mapped to the hg38 genome using the meRanGh tool (align bsRNA-seq reads to reference using HISAT2 v2.1.0) in MeRanTK with options (−fmo). Only uniquely mapped reads were retained and used for m5C sites detection.

The library complexity was analysed for each dataset and represented by PCR Bottlenecking Coefficient 1, calculated as the number of genomic locations where exactly one read maps uniquely divided by the number of distinct genomic locations to which read maps uniquely. For low complexity datasets (T24 cells, [Chen et al, 2019]), PCR duplicates were marked by Picard Mark-Duplicates (v2.7.1) with the default parameters, and removed via samtools (v 1.9) by filtering 1024 flag value.

### m5C site calling and annotation for each dataset

The discovery of m5C sites from each dataset was performed similarly to Schumann et al (2020). Read coverage at each cytosine position in the genome was obtained using the "mpileup" function in samtools (v1.9). The C-to-T mismatches on forward reads and G-to-A mismatches on reverse reads were regarded as unconverted cytosines, and called using a custom script with parameters "-minBQ 30 --overhang 6," where reads were filtered by minimum base quality score 30 and removed terminal 6 nt to avoid overestimation of non-conversion. For ribosome profiling data, data from individual fraction libraries were combined with their respective biological replicate.

Reads containing more than three unconverted cytosines were considered as conversion failure and removed from the bam files ("3C filter"). In the meantime, the RNA icSHAPE values determined by Sun et al (2021) in HeLa, HEK293 and HepG2 cells were downloaded from GEO (GSE145805) to confirm that m5C sites flagged by the "3C filter" were in regions with low icSHAPE values, that is, in highly structured regions. Then candidate sites with a signal-to-noise ratio <0.9 (3C/raw; "S/N ≥ 0.9") were dropped to reduce false positives. To retain high-confidence non-conversion sites, the following criteria were applied: (1) Minimum total read coverage was set as 20 ("20RC"); (2) non-converted C ≥ 3 ("3C"); (3) C + T coverage ≥ 80% ("80CT"); (4) non-conversion of ≥ 10% ("10 MM"); (5). For replicates integration, we select m5C candidates detected in at least two replicates, and set non-converted C ≥ 5 for sites detected in only one replicate. Candidate m5C sites were annotated to gene locus and transcript types according to the hg38 GENCODE v32 annotation (UCSC) using "intersectBed" in bedtools and mapped to six features simultaneously: 5′UTR, CDS, 3′UTR, ncRNA_exonic, intronic, and intergenic.

### Distribution and enrichment of m5C along mRNA

Only m5C sites on exonic protein-coding transcripts were used in this analysis. mRNAs with m5C sites were divided into three segments (5′UTR, CDS, and 3′UTR) and normalised according to their average length. All C positions on the transcripts with m5C candidates were selected as background control. The relative position of each candidate site and background C in the corresponding segment were noted. We identified the relative position of m5C sites and background C around the start codon and stop codon separately to calculate the spatial enrichment for each bin.

### NSUN-enzyme dependence of m5C sites

We collected NSUN2 (Yang et al, 2017; Huang et al, 2019) and NSUN6 depletion datasets (Liu et al, 2021a) for NSUN2/6-dependent m5C sites analysis as described in Chen et al (2019). In detail, m5C sites with more than 0.05 methylation ratio decrease and methylation level reduced to less than 0.1 in NSUN2 or NSUN6 depleted cells were considered to be catalysed by NSUN2 or NSUN6, respectively. In particular, for NSUN2-dependent m5C we used the union set of Yang et al (2017); Huang et al (2019). The sequence of position 0 to +5 of NSUN-dependent m5C sites were then used to calculate position weight matrix (PWM) of consensus motifs. NSUN-dependence of m5C sites in union set can be further predicted by PWM through FIMO (Find Individual Motif Occurrences) in the MEME suite (v 5.4.1). For potential NSUN5-dependent sites, we used NSUN5-overexpression and epigenetically silenced NSUN5 data in LN229

cells (Janin et al, 2019), and regarded sites with 0.05 methylation ratio increase and methylation level higher than 0.1 in NSUN5-OE group as potential NSUN5-dependent sites. All the consensus motifs are plotted by ggseqlogo (Wagih, 2017).

### Proximity to RBP binding sites

RBP footprints reported by Van Nostrand et al (2020) were downloaded from ENCODE; ALYREF footprints in HeLa cells line were obtained from CLIPdb database (Yang et al, 2015) as a positive control. For ENCODE datasets, the intersections of two biological replicates with fold-enrichment ≥4 and $P \leq 10^{-3}$, were filtered as significant peaks and the middle sites of the peaks were regarded as RBP binding sites. Genes with both RBP binding sites and m$^5$C sites were considered for enrichment analysis. Bins were divided around RBP binding sites with the number of m$^5$C sites and total C within the bins. Then Fisher's exact test was applied to calculate the enrichment significance. We set up a gradient region as ±20, ±30, ±50 nt, and ±70 nt, to test the influence of bin size. Finally, ±50 nt was used for further analysis. The relative distance of m$^5$C sites to RBP binding sites was identified and the background C within the same region to get the distribution of m$^5$C around RBP binding sites.

### Cell culture and transfection

HeLa cells (human cervical cancer) were obtained from ATTC and confirmed with CellBank Australia. Cells were grown in DMEM medium (Gibco) supplemented with 10% FBS and 1% antibiotic-antimycotic solution (Sigma-Aldrich) and passaged when 70–90% confluent.

For knockdown experiments, $150 \times 10^3$ cells were plated in 35 mm plates and transfected 6–12 h later with the siRNA against the target selected (FlexiTube siRNA UPF-1, Cat. No. GS5976; QIAGEN) or the negative control (non-silencing siRNA, Cat. No. 1022076; QIAGEN) with a final concentration 30 nM, using 5 $\mu$l of Lipofectamine RNAiMAX Reagent (Thermo Fisher Scientific) and 300 $\mu$l of Opti-MEM (Thermo Fisher Scientific). The medium was replaced 12 h later and cells were harvested 48 h later.

### Protein analysis

Cells were harvested with 200–500 $\mu$l of Protein Extraction Buffer (50 mM Tris pH 7.5, 5 mM EDTA, 150 mM NaCl, 21.5 mM MgCl$_2$, 10% glycerol, 1% Triton, 1X PIC [cOmplete, EDTA-free Protease Inhibitor Cocktail; Sigma-Aldrich]) and incubated 10 min on ice, then incubated on a rotator for 30 min at 4°C and centrifuged at 17,000$g$ for 10 min at 4°C.

The supernatant was transferred to a clean tube, used, or stored at −80°C. Total protein concentration was measured through the Qubit Protein Assay Kit (Thermo Fisher Scientific) following manufacturer's instructions. 30 $\mu$g of proteins were loaded on NuPage 4–12% Bis-Tris Protein Gels (Invitrogen) followed by transfer onto PVDF membrane. The membrane was blocked in Odyssey Blocking Buffer (for IR-Dye detection; LI-COR 927-40000) and probed with primary antibodies: (anti-NSUN2 (1:1,000, 20854-1-AP; Proteintech), anti-NSUN5 (1:1,000, 15449-1-AP; Proteintech), anti-UPF1 (1:1,000, ab86057; Abcam), anti-ACTB (1:1,000, sc-4778 AF790; SantaCruz). The

membranes were probed with a secondary antibody, either anti-mouse-IR-Dye800 (1:10,000, 926-32210; LI-COR) or anti-rabbit-IR-Dye680 (1:10,000, 925-68071; LI-COR), and imaged using the Odyssey CLx Imaging System (LI-COR).

### RNA analysis

Extraction of total RNA was performed using the Direct-zol RNA Miniprep (Zymo Research) kit according to the manufacturer's instructions. Reverse transcription reactions were performed with PrimeScript RT Master Mix (Takara Bio) on 0.5–1 $\mu$g of total RNA in a 10 $\mu$l reaction. RT-qPCR analyses were performed with 20 ng equivalent of cDNA, 5 $\mu$l of 2X SYBR Mastermix (QIAGEN), 1 $\mu$l of 5 $\mu$M primers, and water to a final volume of 10 $\mu$l. DNA amplification (followed by melting curve analysis) was monitored with a QuantStudio 12k Flex Realtime PCR instrument. Primer sequences used for RT-qPCR are listed in Table S4.

Relative RNA quantity was calculated as the fold change (ΔΔCt) with respect to the experimental control sample set as 1 and normalised over ACTB. The ratio of each sample versus its experimental control was tested by two-tailed $t$ test.

### UPF1 knockdown RNA-seq and differential expression analysis

The raw reads were subjected to FastQC (v0.11.9). Low-quality bases and adaptor sequences were removed using Trimmomatic (v0.38) with options (ILLUMINCLIP:Adapter.fa:2:30:10:8:true LEADING:3 TRAILING:3 SLIDINGWINDOW: 4:20 MINLEN:50). Clean reads were mapped to 4S rRNA using bowtie2 (v2.3.5) with parameters ("-q --sensitive --reorder --no-unal --un-conc-gz"). Unmapped reads were mapped to hg38 genome using HISAT2 (v2.1.0) with parameters ("--no-softclip --score-min L,-16,0 --mp 7,7 --rfg 0,7 --rdg 0,7 --max-seeds 20 -k5 --dta"). The mapped reads were processed to featureCounts (v2.0.1) for feature counting.

To quantify RNA, fragment abundance in genes with ≥50 mapped reads were selected and normalised using the size factors estimated by the median of all genes implemented in the DESeq2 (v1.30.1) Bioconductor package. Differential expression analysis was performed by DESeq2. Genes with log$_2$(fold change) ≥ 1.2 and adjusted $P$-value ≤ 0.05 were regarded as differentially expressed genes.

### Cross-linking immunoprecipitation

One million HeLa cells were washed twice with 1X PBS and irradiated once at 150 mJ·cm$^{-2}$ at 254 nm using the Stratalinker UV cross-linker. Cells were then scraped in CLIP Lysis Buffer (50 mM Tris–HCl, pH 7.4, 100 mM NaCl, 1% Igepal CA-630, 0.1% SDS, 0.5% sodium deoxycholate, 1X PIC [cOmplete, EDTA-free Protease Inhibitor Cocktail; Sigma-Aldrich], and RNasin Plus Ribonuclease Inhibitor [Promega]), incubated on ice for 5 min and passed through a 21G needle. Lysates were treated with 3 $\mu$l of TURBO DNase (Thermo Fisher Scientific) for 3 min at 37°C while shaking at 1,100 rpm using a Thermomixer R (Eppendorf). Samples were then centrifuged for 10 min at 17,000$g$ at 4°C to clear the lysate. The supernatant was collected and quantified using Qubit Protein

Assay Kit (Thermo Fisher Scientific) following manufacturer's instructions.

50 μl of Dynabeads Protein G (Thermo Fisher Scientific) were previously washed twice in CLIP Lysis Buffer and then resuspended in the same buffer with either 9 μg of anti-UPF1 goat polyclonal antibody (Cat. No. A300-038A), or 9 μg of Abcam mouse IgG2a Isotype Control antibody (Cat. No. ab18413) in a final volume of 100 μl and incubated in rotation at room temperature for 1 h for antibody conjugation. The conjugated beads were then incubated overnight on a rotator at 4°C with 500 μg of lysate. 50 μg of lysates was also rotated to be used as inputs. The next day beads were washed four times in High-Salt Buffer (50 mM Tris–HCl, pH 7.4, 1 M NaCl, 1 mM EDTA, 1% Igepal CA-630, 0.1% SDS, 0.5% sodium deoxycholate) and twice in PK buffer (100 mM Tris–HCl pH 7.4, 50 mM NaCl, 10 mM EDTA). Beads were resuspended in 50 μl of PK buffer, of which 10 μl were used for Western Blotting analysis and 40 μl were digested with Proteinase K Solution (Cat. No. 25530049; Thermo Fisher Scientific) while shaking at 1,100 rpm in a Thermomixer R (Eppendorf) for 30 min at 37°C for RNA analysis. RNA was purified by phenol/chloroform extraction and precipitated for RT-qPCR analysis.

## Data Availability

All source code used during this study is available at https://github.com/YangLab/Epitranscriptomic-maps-of-5-methylcytosine. All raw and processed sequencing data generated in this study have been submitted to the NCBI Gene Expression Omnibus (GEO; https://www.ncbi.nlm.nih.gov/geo/) under accession number GSE252369.

## Supplementary Information

## Acknowledgements

The authors wish to thank the following Shine-Dalgarno Centre colleagues: Gaetan Burgio for assistance with HeLa knockout cell line generation as well as Eduardo Eyras and Rippei Hayashi for help with resourcing the project. This research was funded in part through grants by the Australian Research Council (DP210102385 to T Preiss, Rippei Hayashi, and Eduardo Eyras; DP180100111 to T Preiss and NE Shirokikh), by the National Health and Medical Research Council of Australia (APP2018363 and APP1135928 to T Preiss and Investigator Grant GNT1175388 to NE Shirokikh), and by National Natural Science Foundation of China (31925011 to L Yang).

### Author Contributions

M Guarnacci: conceptualization, formal analysis, validation, investigation, visualization, methodology, and writing—original draft, review, and editing.

P-H Zhang: data curation, software, formal analysis, investigation, visualization, methodology, and writing—original draft, review, and editing.

M Kanchi: investigation and methodology.

Y-T Hung: investigation and methodology.

H Lin: investigation and methodology.

NE Shirokikh: conceptualization, supervision, funding acquisition, and methodology.

L Yang: formal analysis, supervision, funding acquisition, and writing—original draft.

T Preiss: conceptualization, resources, formal analysis, supervision, funding acquisition, investigation, methodology, project administration, and writing—original draft, review, and editing.

### Conflict of Interest Statement

The authors declare that they have no conflict of interest.

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
