## [Reviewer comments · Life Science Alliance]

Life Science Alliance

Substrate diversity of NSUN enzymes and links of 5-methylcytosine to mRNA translation and turnover

Marco Guarnacci, Pei-Hong Zhang, Madhu Kanchi, Yu-Ting Hung, Hanrong Lin, Nikolay Shirokikh, Li Yang, and Thomas Preiss
DOI: <https://doi.org/10.26508/lsa.202402613>

Corresponding author(s): Thomas Preiss, Australian National University and Thomas Preiss, Australian National University

Review Timeline:

Submission Date:	2024-01-22
Editorial Decision:	2024-03-15
Revision Received:	2024-06-24
Editorial Decision:	2024-06-27
Revision Received:	2024-06-28
Accepted:	2024-06-28

Transaction Report:

March 15, 2024

Re: Life Science Alliance manuscript #LSA-2024-02613-T

Prof. Thomas Preiss
Australian National University
Shine-Dalgarno Centre for RNA Innovation
131 Garran Road
Acton
Canberra, ACT 2601
Australia

Dear Dr. Preiss,

Thank you for submitting your manuscript entitled "Substrate diversity of NSUN enzymes and links of 5-methylcytosine to mRNA translation and turnover" to Life Science Alliance. The manuscript was assessed by expert reviewers, whose comments are appended to this letter. We invite you to submit a revised manuscript addressing the Reviewer comments.

Thank you for this interesting contribution to Life Science Alliance. We are looking forward to receiving your revised manuscript.

Sincerely,

B. MANUSCRIPT ORGANIZATION AND FORMATTING:

Reviewer #1 (Comments to the Authors (Required)):

In this manuscript, Guarnacci et al. reevaluated m5C sites in mRNA reported in 6 different published studies. The authors collected the corresponding bisulfite sequencing data from 7 human tissues and 5 cell lines and subjected them to their previously published (PMID: 32293435) analysis pipeline. Based on this, they defined a union set of 6393 m5C sites, with each sample having several hundred sites. Further processing of the data confirmed findings from previous studies regarding the relative distribution of m5C sites along mRNAs, the sequence logo and the enzymes involved in the modifications. They also examined the overlap of m5C sites with data sets from RNA binding protein (RBP) CLIP analyses. They found 7 RBPs whose binding sites were enriched in the vicinity of m5C sites and selected the mRNA decay factor UPF1 for further experimental analysis. The experiments showed that knock-down of UPF1 resulted in the accumulation of mRNAs with potential m5C sites, whereas this accumulation was lost in a knock-out cell line of the mRNA-specific methyltransferase NSUN2. Curiously, however, the ability of UPF1 to bind to these transcripts increased in the absence of NSUN2.

Even though the manuscript does not provide ground-breaking new data, refining and extracting the most reliable m5C sites from multiple disconnected datasets definitely has its merits and will be of significant value to the scientific community. There are some minor points that should be addressed to strengthen the usefulness and claims of the study.

1. As mentioned above, the availability of a unified dataset of BS-seq analyses that were processed with the same pipeline and reasonably strict parameters, will be quite useful for other researchers. To make it even more so, the authors should provide a table or database, in which all the sites are listed together with the corresponding information (annotation, methylation rate, NSUN2,6,5 target, cell line etc).
2. Fig. 3 and Suppl. Figure 5: It would be interesting to see, which RBPs come up if the analysis is done separately with NSUN2/6/5/others target sites.
3. Fig. 4 and 5: The data suggest that UPF1 promotes RNA degradation when m5C is nearby. Thus, one may speculate that m5C levels should show an increase when UPF1 is not present, because those RNAs would not be removed. This could be tested quite easily on some of the targets for which accumulation is observed in UPF KD cells.
4. Figure 4F seems to show highest significance between Groups I and III and not I and V as written in the text.

Reviewer #2 (Comments to the Authors (Required)):

A major hurdle in the study of many RNA modifications is the frequent lack of consensus between transcriptome-wide maps (whether due to biological context, methodology, or site-calling analysis), often making the establishment of a "ground truth" for studying the biological effects of mRNA modifications difficult. We therefore appreciate the efforts the authors in this manuscript put forth to perform an integrative analysis of human m5C maps to resolve the activity of writers NSUN2/5/6 and identify both previously characterized and novel potential m5C-sensitive RNA binding proteins, and an NSUN2-dependent regulatory phenotype for proposed m5C binder UPF1. While we find this manuscript to be an excellent contribution to the field, before recommending publication we have the following requests:

Major comments:

To confirm that UPF1 binding is truly m5C dependent (and not NSUN2 dependent) we would like to see RNA pulldown experiments on synthesized UPF1 binding sites with or without m5C, showing specific enrichment of UPF1, or another in-vitro demonstration of specificity between m5C and UPF1.

Minor comments:

"Expression levels of m5C methyltransferase enzymes differ substantially across different tissues and cell lines (e.g. NSUN2/5/6 Suppl. Fig. 3B)". What, if any, is the relationship between writer expression and the number or stoichiometry of m5C?

Additionally, please include a plot showing m5C stoichiometry distribution across cell contexts.

"We re-analyzing NSUN2 knockout or knockdown data from Hela cells we found..." it's unclear what this re-analysis entailed - how were NSUN-dependent sites called in this study differently from the studies cited? And what is the justification that this analysis is more accurate / informative?

"It is reported that genes with comparable expression levels in different cell lines show similar RBP enrichment peaks, justifying our approach to compare eCLIP and m5C data from different sources". Given that HepG2 has lower NSUN2 expression and detected m5C sites - we might expect this not to hold since the loss of a modification in HepG2 would lead to the loss of a binding peak. While in this case this discrepancy only leads to a false negative - we would still like this point addressed.

Referee Cross-comments:

I agree strongly with the other reviewer's suggestion that the authors should provide a table or database, in which all the sites are listed together with the corresponding information (annotation, methylation rate, NSUN2,6,5 target, cell line etc).

Australian
National
University

24/06/2024

Thomas Preiss, PhD
Head RNA Biology Group & Director
Shine-Dalgarno Centre for RNA
Innovation; Division of Genome Science
and Cancer / John Curtin School of
Medical Research / College of Health
and Medicine
+61 2 6125 9690
thomas.preiss@anu.edu.au

Point-by-point response to the reviewers' comments:

We thank both reviewers for their generally very positive comments and helpful suggestions for improvement. Below we address each suggestion.

Reviewer #1

1. As mentioned above, the availability of a unified dataset of BS-seq analyses that were processed with the same pipeline and reasonably strict parameters, will be quite useful for other researchers. To make it even more so, the authors should provide a table or database, in which all the sites are listed together with the corresponding information (annotation, methylation rate, NSUN2,6,5 target, cell line etc).

We now provide this information as two new Supplementary Tables 2 and 3.

2. Fig. 3 and Suppl. Figure 5: It would be interesting to see, which RBPs come up if the analysis is done separately with NSUN2/6/5/others target sites.

We now provide these analyses for NSUN2/6/others in this rebuttal.

We generally found our seven RBPs to still show an enrichment around m⁵C mediated by either NSUN2, NSUN6 or other. No clear segregation of RBP enrichment with MTase-specific target sites appears to be evident, therefore we did not include such analyses in the manuscript. However, we noticed how these observed enrichments are more statistically significant when looking at NSUN2-dependent sites, compared to the other two conditions, which is likely due to the generally higher number of m⁵C sites under NSUN2.

3. Fig. 4 and 5: The data suggest that UPF1 promotes RNA degradation when m⁵C is nearby. Thus, one may speculate that m⁵C levels should show an increase when UPF1 is not present, because those RNAs would not be removed. This could be tested quite easily on some of the targets for which accumulation is observed in UPF KD cells.

We performed the suggested UPF1 KD in HeLa cells, followed by bsRNA-Seq. Although the data showed the technical success of the experiment, we realised that the read coverage of the relevant UPF1 footprint-proximal m5C sites was generally well below the numbers required for a quantitative assessment. Thus, to properly address the issue further experimentation and much deeper sequencing will be required. Given the time and expense pressures involved, we had to pause this line work and instead plan to further pursue it as part of a follow-up study.

4. Figure 4F seems to show highest significance between Groups I and III and not I and V as written in the text.

This was not expressed very clearly in the original submission. What we meant to say is that Group V shows the highest difference in RNA level (0.11) compared to Group I (0) – Group III = 0.03. The wording was corrected in the main manuscript text on page 9.

Reviewer #2

Major comments:

1. To confirm that UPF1 binding is truly m5C dependent (and not NSUN2 dependent) we would like to see RNA pulldown experiments on synthesized UPF1 binding sites with or without m5C, showing specific enrichment of UPF1, or another in-vitro demonstration of specificity between m5C and UPF1.

This is a great suggestion, but we feel that it would go beyond the scope of the current study. Instead, we plan to perform such experiments as part of a follow-up study.

Minor comments:

1. "Expression levels of m5C methyltransferase enzymes differ substantially across different tissues and cell lines (e.g. NSUN2/5/6 Suppl. Fig. 3B)". What, if any, is the relationship between writer expression and the number or stoichiometry of m5C?

This kind of relationship has not been explored much thus far. Attempting to address this across different studies is further complicated by technical differences such as sequencing depth, bioinformatic pipeline and others. Nevertheless, we have looked at this within the confines of our union set and reported our findings in the new Supplementary Figure 3, as well as in Figure 2D.

Broadly, we found that the stoichiometry of m5C sites seemed to be unaffected by changes in NSUNs expression level, with the non-conversion level being consistent around 20% across all samples (new Supplementary Figure 3E). On the other end, the number of modifications detected (new Supplementary Figure 3C) generally correlates with read coverage (new Supplementary Figure 3B), having deeply sequenced samples with higher number of m5C sites called and vice versa.

Interestingly, in our new Figure 2D we show a significant correlation between NSUN2 or NSUN6 expression level and the methylation level of corresponding m5C sites. Further, the methylation level of sites that were called as NSUN2/6-independent ("Other sites") had no apparent correlation with the expression level of either MTase.

2. Additionally, please include a plot showing m5C stoichiometry distribution across cell contexts.

We now show violin plots of m5C stoichiometry levels for all datasets as a new panel for Supplementary Figure 3E.

3. "By re-analyzing NSUN2 knockout or knockdown data from Hela cells we found..." it's unclear what this re-analysis entailed - how were NSUN-dependent sites called in this study differently from the studies cited? And what is the justification that this analysis is more accurate / informative?

Among the studies referenced in this manuscript there is a consensus around what filters and criteria need to be met for a candidate m5C site to qualify as high confidence. However, there are instances where different

levels of stringency are applied in different studies. NSUN dependency in this study was calculated following the method described in Chen et al, 2019. Briefly, sites were calculated as differentially methylated when they showed a methylation difference ≥ 0.05 and a reduction of their methylation level to less than 0.1 upon MTase knockdown or knockout. Details of such method are described in “A union list of m⁵C sites in human transcriptomes” in the results section, as well as in “m⁵C site calling and annotation for each dataset” and “NSUN-enzyme dependence of m⁵C sites” in methods.

The main difference in m⁵C sites calling between Huang et al., 2019 and our analysis is the following: in Huang et al., 2019, a candidate site was called as methylated even when it reached the 10% methylation cut-off in only one of the two replicates, if it could pass the more stringent 5C filter (instead of 3C) and if the site had RC < 20 in the other replicate. While in our pipeline we used the 5C filter (instead of 3C) when a site was methylated > 10% in only one of the two replicates, regardless of the read coverage of the same site in the other replicate.

Also, our results indicated the binomial test used in Huang’s study had minimal impact on m⁵C detection – as also indicated in their Supplementary Figure 4E (Huang et al, 2019). Therefore, we decided not to use this statistical test in the final version of our study. Finally, we used different mapping software and custom scripts, which could also lead to minor differences in results.

As a result, we obtained more NSUN2-dependent sites when re-analysing data from Huang et al., 2019 compared to their original analysis (A), as we were able to assess m⁵C sites with lower read coverage (B) and lower (median) methylation level (C). Finally, our re-analysis allowed the mapping of NSUN2 sites also on normally underrepresented genomic location, such as introns and intergenic (E), despite virtually identical sequence logo (D).

(A) Venn diagram showing the overlap of NSUN2-dependent m⁵C sites calculated in WT HeLa cells in Huang et al, 2019 and this study. (B) Read coverage, (C) non-conversion ratio, (D) consensus sequence and (E) genomic distribution of "overlap" m⁵C and "this study only" m⁵C sites.

- "It is reported that genes with comparable expression levels in different cell lines show similar RBP enrichment peaks, justifying our approach to compare eCLIP and m⁵C data from different sources". Given that HepG2 has lower NSUN2 expression and detected m⁵C sites - we might expect this not to hold since the loss of a modification in HepG2 would lead to the loss of a binding peak. While in this case this discrepancy only leads to a false negative - we would still like this point addressed.

This is a good point although it is not necessarily a given that RBPs binding sites would be entirely lost if an m5C site nearby is absent. This would happen if the RBP in question is an m5C reader in the strict sense, e.g., will not bind unless m5C is present. The reality for direct m5C readers and particularly for RBPs whose function might be affected by m5C nearby likely is that their binding might be more gradually affected. We have a brief qualification to the main manuscript text on page 11.

Referee Cross-comments:

I agree strongly with the other reviewer's suggestion that the authors should provide a table or database, in which all the sites are listed together with the corresponding information (annotation, methylation rate, NSUN2,6,5 target, cell line etc).

This was done as explained in our reply to reviewer #1.

June 27, 2024

RE: Life Science Alliance Manuscript #LSA-2024-02613-TR

Prof. Thomas Preiss
Australian National University
Shine-Dalgarno Centre for RNA Innovation
131 Garran Road
Acton
Canberra, ACT 2601
Australia

Dear Dr. Preiss,

Thank you for submitting your revised manuscript entitled "Substrate diversity of NSUN enzymes and links of 5-methylcytosine to mRNA translation and turnover". We would be happy to publish your paper in Life Science Alliance pending final revisions necessary to meet our formatting guidelines.

- please be sure that the authorship listing and order is correct
- please add a callout for Figure S3E to your main manuscript text

FIGURE CHECKS:

- please add sizes next to blots in Figures 5A and 6A

A. FINAL FILES:

B. MANUSCRIPT ORGANIZATION AND FORMATTING:

Sincerely,

June 28, 2024

RE: Life Science Alliance Manuscript #LSA-2024-02613-TRR

Prof. Thomas Preiss
Australian National University
Shine-Dalgarno Centre for RNA Innovation
131 Garran Road
Acton
Canberra, ACT 2601
Australia

Dear Dr. Preiss,

Thank you for submitting your Research Article entitled "Substrate diversity of NSUN enzymes and links of 5-methylcytosine to mRNA translation and turnover". It is a pleasure to let you know that your manuscript is now accepted for publication in Life Science Alliance. Congratulations on this interesting work.

DISTRIBUTION OF MATERIALS:

Again, congratulations on a very nice paper. I hope you found the review process to be constructive and are pleased with how the manuscript was handled editorially. We look forward to future exciting submissions from your lab.

Sincerely,
